# Intracellular Fate of Sub-Toxic Concentration of Functionalized Selenium Nanoparticles in Aggressive Prostate Cancer Cells

**DOI:** 10.3390/nano13232999

**Published:** 2023-11-22

**Authors:** Caroline Bissardon, Olivier Proux, Salvatore Andrea Gazze, Odile Filhol, Benoît Toubhans, Lucie Sauzéat, Sylvain Bouchet, Aled R. Lewis, Thierry Maffeis, Jean-Louis Hazemann, Sam Bayat, Peter Cloetens, R. Steven Conlan, Laurent Charlet, Sylvain Bohic

**Affiliations:** 1Université Grenoble Alpes, Inserm, UA7, Synchrotron Radiation for Biomedicine (STROBE), 38400 Grenoble, France; sbayat@chu-grenoble.fr; 2OSUG, UAR 832 CNRS, Université Grenoble Alpes, IRD, INRAe, Météo-France, 38041 Grenoble, France; proux@esrf.fr; 3Swansea University Medical School, Swansea University, Swansea SA2 8PP, UK; s.a.gazze@swansea.ac.uk (S.A.G.); benoit.toubhans@gmail.com (B.T.); a.r.lewis@swansea.ac.uk (A.R.L.); t.g.g.maffeis@swansea.ac.uk (T.M.); r.s.conlan@swansea.ac.uk (R.S.C.); 4Laboratoire de Biologie et Biotechnologies pour la Santé, IRIG-DS, Inserm U1292, CEA, Université Grenoble-Alpes, 38054 Grenoble, France; odile.filhol-cochet@cea.fr; 5Institute of Biogeochemistry and Pollutant Dynamics, ETH Zurich, CH-8092 Zurich, Switzerland; lucie.sauzeat@uca.fr (L.S.);; 6Institut Néel CNRS-UGA, 25 Avenue des Martyrs, 38042 Grenoble, France; hazemann@esrf.fr; 7ESRF—The European Synchrotron Radiation Facility, 38043 Grenoble, France; cloetens@esrf.fr; 8ISTerre, Université Grenoble Alpes, 38041 Grenoble, France

**Keywords:** prostate cancer, selenium, nanoparticle-correlative synchrotron imaging, X-ray fluorescence, speciation, cytotoxicity

## Abstract

Selenium 0 (Se^0^) is a powerful anti-proliferative agent in cancer research. We investigated the impact of sub-toxic concentrations of Se^0^ functionalized nanoparticles (SeNPs) on prostate cancer PC-3 cells and determined their intracellular localization and fate. An in-depth characterization of functionalized selenium nanoparticles composition is proposed to certify that no chemical bias relative to synthesis issues might have impacted the study. Selenium is an extremely diluted element in the biological environment and therefore requires high-performance techniques with a very low detection limit and high spatial resolution for intracellular imaging. This was explored with state-of-the-art techniques, but also with cryopreparation to preserve the chemical and structural integrity of the cells for spatially resolved and speciation techniques. Monodisperse solutions of SeNPs capped with bovine serum albumin (BSA) were shown to slow down the migration capacity of aggressive prostate cancer cells compared to polydisperse solutions of SeNPs capped with chitosan. BSA coating could prevent interactions between the reactive surface of the nanoparticles and the plasma membrane, mitigating the generation of reactive oxygen species. The intracellular localization showed interaction with mitochondria and also a localization in the lysosome-related organelle. The SeNPs-BSA localization in mitochondria constitute a possible explanation for our result showing a very significant dampening of the PC-3 cell proliferation capabilities. The purpose of the use of sublethal compound concentrations was to limit adverse effects resulting from high cell death to best evaluate some cellular changes and the fate of these SeNPs on PC-3. Our findings provide new insight to further study the various mechanisms of cytotoxicity of SeNPs.

## 1. Introduction

Selenium (Se) is fundamental to human health. It is part of the active site of glutathione peroxidase and thioredoxin reductase or deiodinase enzymes and a potent antioxidant agent with evidence of anticancer activity [1,2,3]. A number of studies provide compelling evidence that Se can be an effective chemopreventive and chemotherapeutic agent [1], and this fact has been particularly supported by epidemiological, preclinical, and clinical studies [4]. The bioavailability and side effects of Se were clearly shown to be closely related to its chemical species [5]. Still, one of the limitations of the use of organic or inorganic forms of Se, such as sodium selenite (Na_2_SeO_3_), sodium selenate (Na_2_SeO_4_), or selenomethionine (SeMet), is their narrow therapeutic window, impeding their translation into the clinic.

The potential of low-toxicity and novel therapeutic properties offered by selenium nanoparticles (SeNPs) and their high bioavailability have attracted attention and are expected to have a higher therapeutic index compared to other forms of inorganic and organic selenium compounds [6,7,8]. Chemically synthesized SeNPs have been studied as potent chemotherapeutic agents or as drug carriers, also capped with some polysaccharides used as reducing and stabilizing agents within the synthesis of SeNPs. The latter were shown to both significantly inhibit the growth of cancer cells as well as exhibit interesting in vitro and in vivo antioxidant activities by activating selenoenzymes [8,9,10,11,12,13]. The functionalization of SeNPs is becoming a popular strategy in order to overcome the above-mentioned limitation and improving the bioactivity of SeNPs [14]. The use of bovine serum albumin (BSA) or chitosan is of interest in the potential application of SeNPs as anticancer agents.

Bovine serum albumin (BSA), a highly water-soluble and abundant carrier protein of blood plasma extensively studied and sharing high homology with human serum albumin, received considerable attention for the development of new nanomedicine [15]. The modification of the surfaces of nanoparticles with proteins such as BSA, a cost-effective solution, prevents nanoparticles from agglomerating and provides low immunogenicity, enhanced biocompatibility, and some added functionalities using interaction or coupling with organic and inorganic ligands through amine and sulfhydryl groups of albumins [16]. Furthermore, if albumin is known to strongly associate with inorganic nanoparticles [15], it also contributes to some preferential accumulation in tumor and tumor cells through some receptors such as glycoprotein receptor Gp60 or Secreted Protein Acidic Rich in Cysteine (SPARC, also known as osteonectin; or basement-membrane-40, BM-40), whose overexpression is often associated with tumor growth, metastasis, and aggressiveness [17,18], a multifunctional glycoprotein that modulates the interaction of cells with the extracellular environment [19]. We previously reported the effects of SeNPs-BSA and SeNPs-Chitosan on ovarian cancer cells and demonstrated that both are effective in inhibiting their growth and decreasing their metastatic potential, associated with the altered nanomechanical response of the cell membrane [20]. Chitosan is an interesting non-toxic positively charged polysaccharides that have a low-immunogenicity and a high biodegradability and is a biocompatible polysaccharide used in many fields such as tissue engineering or the delivery of drugs and vaccines [21,22]. Like BSA, it has been also used as a stabilizing or capping agent [23,24]. Well-controlled (20–60 nm), highly stable, and soluble nano red elemental Se has been reported by adding BSA to the redox system of sodium selenite and glutathione [25]. Similarly, SeNPs-Chitosan has been obtained, resulting in positively charged nanoparticles that provide some selectivity towards cancer cells compared to healthy human cells and enhance their cellular uptake and induce cytotoxicity in human melanoma cancer cells through cell apoptosis [26] or in human hepatocarcinoma (HepG2) cells [9].

SeNPs functionalized with BSA have been shown to reduce, by seven-fold, acute toxicity in mice compared to sodium selenite. Interestingly, this type of nanoparticle has anticancer properties and suppresses LNCaP prostate cancer cell growth partially through caspases-mediated apoptosis. The androgen-independent PC-3 prostate cancer cells derived from bone metastases represent a far more aggressive phenotype than androgen-dependent LNCaP cells derived from human lymph node metastatic lesions of prostatic adenocarcinoma [27]. The effects of BSA- or chitosan-functionalized SeNPs on aggressive prostate cancer cells are still elusive while the intracellular fate of these nanoparticles has remained unexplored. Moreover, it is well established that selenium compounds have anticancer capabilities that correlate with their chemical form and doses, with selenium having four valence states: selenate (Se^VI+^), selenite (Se^IV+^), selenide (Se^II−^), and elemental selenium (Se^0^). Herein, sub-toxic concentrations of SeNPs-BSA and SeNPs-Chitosan were used and compared to sodium selenite salts in PC-3 cancer cells. The results show that SeNPs-Chitosan solution is polydisperse and can make aggregates that influence, to a very large extent, their cellular uptake, contrary to the very stable solution of SeNPs-BSA. We could evidence the presence in cells of redox-active selenium compounds (selenite, selenodiglutatione, and the diselenide type of compound RSeSeR) that generate selenide under reduction but also can generate superoxide in the presence of thiols and oxygen, resulting in the formation of various ROS. Still, the production of selenite in the preparation or in the culture media during incubation cannot be eluded and may contribute to some of the cellular effects of these SeNPs. The cellular uptake of SeNP-BSA is higher than that of SeNP-Chitosan; however, the IC20 of SeNP-BSA is higher than the IC20 of SeNP-Chitosan, with the later having a particle size not optimal for efficient cellular uptake by cancer cells, and the BSA could enhance the uptake of SeNPs-BSA with NPs within optimal sizes (30–50 nm in average), but, also, BSA coating could prevent interactions between the reactive surface of the nanoparticles and the plasma membrane and mitigate the generation of reactive oxygen species. The intracellular localization shows interaction with mitochondria and is also localized in the lysosome-related organelle. The SeNPs-BSA localization in mitochondria is a possible explanation for our result showing a very significant dampening of the PC-3 cell proliferation capabilities. The purpose of the use of sublethal compounds concentrations was to limit adverse effects resulting from high cell death to best evaluate some cellular changes and the fate of these SeNPs on PC-3.

## 2. Materials and Methods

### 2.1. Selenium Nanoparticles

SeNPs were synthetized by a USA company, NANOCS, following protocols defined in the literature by Yu et al. (2012) [26]. SeNPs are made from a mixture of sodium selenite and ascorbic acid (used as a reductant). After synthesis, SeNPs were capped by either bovine serum albumin (BSA) or chitosan used as a coating, inhibiting crystal growth and facilitating cell incorporation. The routine production allowed us to work with reproducible SeNP solutions in order to obtain monodispersed, rather than aggregated, nanoparticles in the culture growth medium. All experiments were carried out with the same lot of monodispersed SeNPs solutions.

### 2.2. Preparation of Working Concentration (Inhibition Concentrations—IC)

SeNP nanoparticles capped with BSA and chitosan, i.e., SeNPs-BSA and SeNPs-Chitosan, respectively (stock solution 2 mg/mL of selenium, NANOCS, New York, NY, USA), were placed in an ultrasound bath for 30 min at room temperature. BSA and chitosan are NP stability agents. Then, the preparation of final working concentrations was obtained through serial dilution. Between each dilution, the solution was slowly vortexed for 1 min followed by pipetting several times to homogenize the solution.

### 2.3. Cell Culture and Treatments

High-metastatic-potential human prostate cancer cell line (PC-3, ECCAC) was used and regularly cultured in ATCC-modified RPMI 1640 medium supplemented with 10% of fetal bovine serum and 1% Penicillin-Streptomycin. Cells were then exposed, or not exposed, to IC20 (80% cell viability) inhibition concentrations (IC in µg/mL of selenium) of SeNPs-BSA or chitosan for 24 h. As a positive control, we used a soluble salt of Se (sodium selenite, Sigma-Aldrich, St. Louis, MO, USA) as a 2 mg/mL selenium stock solution in MilliQ water.

### 2.4. Characterization of Selenium Nanoparticles

#### 2.4.1. Transmission Electron Microscopy (TEM)

Negative-stain-on-grid technique was used for transmission electron microscopy (TEM). Stock of aqueous SeNPs solution was placed in an ultrasound bath at low frequency and low power for 30 min in order to have a monodispersed solution of NPs. Then, 10 µL of the aqueous solution of SeNPs-BSA or chitosan (final concentration of 1 mg/mL after dilution in MilliQ water) was added to a grid coated with a carbon supporting film for 1 min. Further, SeNPs-BSA or -Chitosan (final concentration of 1 mg/mL in cell culture medium) was incubated for 24 h in cell culture medium at 37 °C and 10 µL of solution was added to a grid coated with a carbon supporting film for 1 min. The excess solution was soaked off by a filter paper and the grid was air-dried. No stain was added to the grid. TEM images were taken under low-dose conditions (<10 e^−^/Å^2^) at magnifications of 1.4K×, 13K×, 23K×, and 49K× times, with defocus values between 1.2 and 2.5 μm on a Tecnai 12 LaB6 electron microscope at 120 kV accelerating voltage using CCD Camera Gatan Orius 1000.

#### 2.4.2. X-ray Photoelectron Spectroscopy (XPS)

The samples were prepared for X-ray Photoelectron Spectroscopy (XPS) by drop casting a small amount (<10 μL) of suspension on top of a clean piece of silicon wafer. The samples were then loaded directly into the XPS load lock. XPS scans were acquired using Al Kα monochromatic X-rays at 1486.6 eV. The pass energy was 160 eV for survey scans and 20 eV for core-level scans. In an effort to increase the signal from SeNPs, the samples were etched in situ with a 2 × 2 mm^2^ raster of Ar500+ clusters at 5 kV for 5 min before scanning a second time. Etching with Ar clusters is known to be less aggressive than monoatomic Ar etching and limits both organic sample degradation and the chemical reduction of surface species, although it does not stop them entirely.

#### 2.4.3. X-ray Absorption Near-Edge Spectroscopy (XANES)

XANES measurements were performed at the Se K-edge on the French CRG FAME-UHD beamline of the European Synchrotron Radiation Facility (ESRF, Grenoble, France). The main optical elements of the beamline are a Si(220) two-crystal monochromator surrounded by two Rh-coated silicon mirrors. Horizontal focusing by the 2nd crystal of the monochromator linked to the vertical one by the 2nd mirror allows to have beam size of around 200 × 100 µm^2^ (Full-Width Half-Maximum value) on the sample. Experiments were performed using a liquid helium cryostat, with a sample temperature around 10K during acquisition, in order to limit, at the maximum, any radiation damages. Selenium concentration being very low, XANES acquisitions were performed in fluorescence mode using a high-energy-resolution Crystal Analyzer Spectrometer (CAS) in the Johann geometry, equipped with 5 spherically bent Ge(844) crystals (radius of curvature: 1 m) optimized to detect the Se-Kα1 fluorescence line (11.222 keV). Such kind of detection system is very well adapted to the study of ultra-high diluted compounds [28]. Se-Kα1 fluorescence photons selected by the CAS were then collected on a silicon-drift detector (energy resolution: ~200 eV) in order to discriminate the signal of interest (diffracted by the CAS) from the background (scattered by the CAS) and so improve the signal-to-background ratio. The total energy resolution of the fluorescence detection, including both the beamline and the spectrometer contributions, was measured at around 3 eV (pseudo-elastic peak measurement). XANES acquisitions were then performed in High-Energy-Resolution Fluorescence Detection (HERFD) mode.

Sample preparation followed a protocol developed for biological samples and selenium references already published [29]. XANES calculations were performed using the FDMNES code [30]. Firstly, the three previously described possible structures for Se^0^ were used as structural input parameters. Different calculation cluster sizes (defined by the radius R) around the central atom were used (3, 4, 5, and 6 Å) in order to mimic both the effect of a limited size and different crystallinity of the nanoparticles (Appendix A). Secondly, absorption cross-sections were ab initio calculated using Finite Difference Method and self-consistent-field theories. Combination of these two methods allows to avoid the limitations imposed by the Muffin Tin approximation and to determine, unambiguously, the Fermi level, which is of particular importance for the absorption edge position. Finally, the raw calculated absorption cross-sections were convoluted with a Lorentzian function in order to take into account the limited core-hole lifetime broadening of the probed edge (2.33 eV for the Se K-edge) [31].

### 2.5. Synchrotron X-ray Fluorescence (XRF) Nanoimaging

PC-3 cells were plated on Si_3_N_4_ membranes (1.5 mm × 1.5 mm membrane size and 500 nm thickness, Silson Ltd., Southam, UK) and incubated for 24 h in regular culture medium with SeNPs-BSA or -Chitosan or sodium selenite at IC20 concentration. Briefly, samples were quickly rinsed in ammonium acetate buffer followed by cryo-fixation in ethane chilled with liquid nitrogen and further freeze-dried at low temperature under vacuum for samples to be analyzed at the hard X-ray Nanoprobe beamline, I14, at Diamond Light Source. Frozen hydrated cells could be kept and analyzed in their near-native state at the hard X-ray cryo-nanoprobe ID16A beamline (ESRF, Grenoble, France).

Experiments on freeze-dried cells were performed at room temperature in air at the hard X-ray Nanoprobe beamline, I14, at Diamond Light Source [32]. Si_3_N_4_ membranes were loaded into the sample holder available at I14, using fine tweezers, with the flat side on the surface (cells facing upwards). An incident energy beam of 14 keV was used, focusing the hard X-ray beam size of 200 nm with a photon flux of ~10^9^ ph/s. Synchrotron-XRF elemental maps were obtained via raster-scanning using different step sizes and XRF spectra were collected using a 4-elements silicon drift detector (SGX-RaySpec, High Wycombe, UK) in the backscatter geometry (solid angle ~0.7 sr), with pulse processing and dead-time correction performed with the xspress3 system. Coarse scans were first run to confirm the positions of cells (2 µm step size, 0.1 s) before fine scans (200 nm step size, 0.5 s) were used to analyze elemental distribution with higher resolution.

Frozen–hydrated cells were loaded through cryo-transfer system available at cryo-nanoprobe ID16A with samples kept, at all times, at around 120K and analyzed under a vacuum of 1 × 10^−7^ mbar. A pair of multilayer-coated Kirkpatrick Baez mirrors that are 185 m down-stream of the undulator source was used to focus the incident X-ray at the energy of 17 keV. Cells were raster-scanned with a beam focal spot of 43 × 50 nm^2^, a dwell time of 50 ms, and a step size of 50 nm with a flux of 2.2 × 10^+11^ ph/s. Each sample was placed in normal incidence and two 6-element silicon drift detectors (SGX Sensortech, Whitstable, UK) were aligned within the X-ray focal plane and located perpendicular to the beam path on each side of the sample to collect the fluorescence signals. The summed spectra from the multielement detectors were fitted and analyzed using PyMCa 5.9.1 version of the software developed by the ESRF [33]. The elemental areal mass concentration was calculated using the Fundamental Parameters (FP) approach implemented in PyMca software package. A reference material containing elements of certified concentration (RF7-200-S2371 from AXO, Dresden, Germany) with nanoscale uniform mass depositions (ng/mm^2^ range) was used. The resulting elemental areal mass density maps were visualized with ImageJ 1.54f version of the software.

### 2.6. Transmission Electron Microscopy (TEM) Ultrastructural Analysis of the SeNP Internalization in PC-3 Cells

Cells were collected and cryo-fixed using high-pressure freezing (HPM100, Leica, Wetzlar, Germany) followed by freeze-substitution (EM ASF2, Leica). The freeze substitution procedure uses a mixture of 1% (*w*/*v*) osmium tetroxide in dried acetone and a programmed protocol starting from −90 °C for 60–80 h, followed by a heating rate of 2 °C/h to reach −60 °C, then 10–12 h at −60 °C, followed by a heating rate of 2 °C /h to reach −30 °C, and finally, 10–12 h at −30 °C, quickly heated to 0 °C for 1 h to enhance the staining efficiency of osmium tetroxide and uranyl acetate. The cells were brought back at −30 °C. Then, they were washed four times in anhydrous acetone for 15 min each and gradually embedded in anhydrous araldite resin. Ultrathin sections of 70 nm thickness were mounted onto copper grids coated with formvar and carbon. Sections were then stained in 1% uranyl acetate and lead citrate. Micrographs were obtained using a Tecnai G2 Spirit BioTwin microscope (FEI) operating at 120 kV with an Orius SC1000 CCD camera (Gatan, Pleasanton, CA, USA).

### 2.7. Determination of the Cellular Se Concentration and Speciation Analyses

All chemical analyses were carried out in clean laminar flow hoods using distilled acids to avoid any exogenous contaminations. For Se concentration measurements, culture media (liquid samples) and cell pellets (solid samples) were first weighted and then digested with a mixture of 2 mL of 15 M HNO^3^ and 1 mL of H_2_O_2_ (30%) in Teflon tubes using a microwave digestion system (CEM, Explorer SP-D 24/48). The digestion program consists of 8 min heating up to 100 °C, 5 min at 100 °C, 10 min heating up to 220 °C, and 5 min at 220 °C, with a maximum pressure of 150 bar. Once digested, solutions were first evaporated at 80 °C to avoid Se losses through evaporation and then diluted in low-concentrated nitric solution (HNO_3_ 0.5N). Se concentrations were then measured with a ICP-MS (Agilent 8800 ICP-MS/MS, Santa Clara, CA, USA) at ETH Zürich using the hydrogen (H^2^) gas mode collision cell (H^2^ gas: 5 mL/min) coupled with Ni cones and micromist^TM^ nebulizer (flow: 0.5 mL/min). The concentrations were calculated using calibration curves. Calibration of the signal was performed using a blank and eight different dilutions of a mixed solution of the TraceCERT^®^ multi-element standard solution V and pure elements (S, Br, Se and As). Lutetium (Lu) and yttrium (Y) (0.1 ppm) were used as internal standards to correct instrumental biases. The accuracy and precision of Se concentrations was assessed based on complete duplicate and re-run analyses of samples, as well as the simultaneous analysis of selenium-enriched yeast-certified reference material (SELM-1). Our measured values are within uncertainty of previously published results and reproducibility is, on average, better than 5% (2σ). Based on the analysis of reference standards and re-run analyses, we therefore estimate that the measurement precision is, on average, better than 5% for Se concentrations. Concentrations are reported in ppb (ng/g) for the culture media samples and in ng/cells for cell pellets in Appendix A. Frozen–hydrated PC-3 cell pellets used for speciation in HERFD-XAS were prepared as detailed in [29]. Briefly, PC-3 cells incubated with selenium compounds for 24 h were thoroughly rinsed with warmed PBS (37 °C) and collected through gentle scraping using a cell scraper. The cell pellet was obtained through centrifugation of the Eppendorf tube and supernatant was discarded. The tube was then flash-frozen in LN^2^ and the pellet was quickly extracted under a glovebox fully purged with an inert atmosphere and next transferred to a manual hydraulic press in LN_2_-cooled mold. Then pellet died. This resulted in a frozen cell pellet of 3 mm diameter that could be transferred, still under the glove box in a cryotube, for storage. These cell cryo-pellets could be further transferred to a 77K pre-cooled cryostat sample holder used on the CRG-FAME-UHD beamline for HERFD-XAS at 10K.

### 2.8. Atomic Force Microscopy (AFM)

Nanowizard II AFM (JPK, Berlin, Germany) mounted on a ZEISS 510 confocal microscope (Zeiss, Cambridge, UK) was used to obtain the force-indentation curves. During AFM, cells were kept alive in a pH indicator-free and serum-free culture medium at 37 °C in a petri dish during analysis lasting 3 h at maximum. The inverted optical microscope was used to position the tip on the cell and force volume, conducted using borosilicate colloidal (Novascan, Oxford, UK) cantilevers, with a nominal spring constant of 0.35 N/m and a radius of 2.5 μm. JPK Data Processing program (https://jpk.software/, accessed on 1 October 2020 was used to process the acquired force curves as detailed in [22]. In order to resolve the membrane architecture, cells were fixed for 30 min in 4% PFA (Merck, Feltham, UK) diluted in PBS at room temperature prior to imaging. PC-3 cell morphology and topography were analyzed using a BioScope Catalyst (Bruker Instruments, Billerica, MA, USA) mounted on a Nikon Eclipse Ti-S inverted optical microscope (Nikon Instruments, Leiden, The Netherlands). The tip was carefully positioned on the cell of interest using the inverted optical microscope and tapping mode imaging was obtained using MLCT-E silicon nitride cantilevers (Bruker-Nano, Coventry, UK).

### 2.9. Cell Viability Assay

The cell cytotoxicity of SeNPs-BSA or -Chitosan or sodium selenite was achieved via the MTT assay (3-(4,5 dimethylthiazol-2-yl)-2,5-diphenyltetrazolium bromide). PC-3 cell (2 × 10^4^ cells/well) were seeded in 96-well plates, and they were left for 48 h in total with a medium change at 24 h. Cells were then exposed to different concentrations of SeNPs-BSA or -Chitosan (range: from 75 to 0.39 μg/mL; serial dilutions (dilution factor 2) [75, 25, 12.5, 6.25, 3.125, 1.5625, 0.78, 0.39 μg/mL]) for a 24 h period. Sodium selenite aqueous solution was used as positive control with the same concentration range from 10 to 0.1 µg/mL. After treatment period, cells were washed twice with PBS, and then, were allowed to react with MTT solution medium (used at a concentration of 500 µg/mL) for a period of 3 h in dark at 37 °C. At the end of the incubation period, dark-purple formazan crystals were formed. These crystals were then washed once with PBS and then were solubilized with an organic solvent—DMSO—and the absorbance at 595 nm was measured spectrophotometrically. This test was performed for a minimum of N = 3 (each N was performed in triplicates). To determine the cell viability, we calculated the percent viability as % viability = [(Optical density {OD} of treated cell − OD of blank)/(OD of vehicle control − OD of blank) × 100].

### 2.10. Wound Scratch Analysis

Cell migration was assessed using well-established wound scratch assay. PC3 cells (25,000 cells in 200 µL of fresh culture medium) were seeded in 96-well plates (IncuCyte^®^ ImageLock Plates, Essen Bioscience, Ann Arbor, MI, USA, 96-well microtiter plates Cat. No. 4379) for 48 h at 37 °C until they reached more than 80% confluence. The plates were placed in an Incucyte wound maker (Essen Bioscience, Ann Arbor, MI, USA) to ensure consistency in the wounds made on the cell monolayers in each well. The medium and cell debris were then removed and replaced with 200 μL of fresh medium containing SeNPs-BSA or -Chitosan or sodium selenite and controls. Then, the plates were placed in the Incucyte Zoom, and the wound width was monitored over 24 h at 37 °C for later analysis using the software incorporated into the IncuCyte Zoom.

### 2.11. Statistical Analysis

Data were presented as means ± standard deviations (SDs). The statistical significance was analyzed with GraphPad Prism 8.0.2 software and statistical tests used are mentioned when appropriate in figure legends. A value of *p* < 0.05 was considered significant.

## 3. Results and Discussion

### 3.1. Physical and Chemical Characterization of the Manufactured Selenium Nanoparticles

#### 3.1.1. Transmission Electron Microscopy

Transmission electron microscopy studies were performed to obtain information on the round shape of the different SeNPs used herein. An average size of 30 nm was evaluated for both nanoparticles and is coherent with the literature. After a sonication of 30 min, NPs were then well individualized. BSA-coated Se-NPs (32.6 ± 12.7 nm) presented a spherical and homogeneous form contrary to chitosan-coated SeNPs (28.3 ± 11.1 nm), which were less homogeneously spherical and tended to form aggregates with increasing concentrations (Figure 1). Indeed, some Dynamic Light Scattering and Zeta Potential measurements were conducted using a ZetaSizer instrument (Malvern Instruments, Malvern, UK). The polydispersity index supported the above findings with a value of 0.220 ± 0.012 for SeNPs-chitosan, which was considered polydisperse, while for SeNPs-BSA, the PDI was found to be 0.123 ± 0.002, which was considered monodisperse. Indeed, SeNP-chitosan had a zeta-potential of 16.4 ± 4.4 mV, which was within the –30 mV and +30 mV range known to indicate the poor stability of the nanoparticles and, very likely, aggregation or agglomeration, while the value of −51.2 ± 15.8 mV for SeNPs-BSA indicated the very high stability of the solution. We noticed that the average hydrodynamic size of SeNPs-BSA was 108 ± 30 nm, and it was 320 ± 221 nm for SeNPs-chitosan; this value with the ZetaSizer was higher than the supplier’s specification, while the very precise measurements through the series of TEM images confirmed those specifications. This can be explained by the fact that the SeNP solution is polydisperse and the presence of bigger particles could contribute to shift the measured particle sizes towards larger values through an increase light scattering. Previous studies have shown that the 30–50 nm sizes of NPs are optimal for intracellular uptake through efficient interaction with the cellular plasma membrane receptors followed by receptor-mediated endocytosis internalization [34]. Of note, larger NPs (>250 nm) were described to be optimal for in vitro phagocytosis, and NPs with sizes in the range of 120–200 nm were described to be mostly uptaken by cells via clathrin- or caveolin-mediated endocytosis [35,36].

#### 3.1.2. X-ray Photoelectron Spectroscopy

XPS being a surface-sensitive technique, measurements were performed in order to precisely characterize the coatings on both BSA- and chitosan-manufactured nanoparticles. The survey scans revealed C, O, Na, N, and F as the main constituents of the surface, with small amounts of Se and P. Si from the substrate was also visible (at 100 eV and 150 eV), especially for the chitosan sample (Figure 2). There could have been some S as well, but because the Se_3s_ line overlapped with the S_2s_ and the Se_3p_ line overlapped with S_2p_, this was hard to ascertain.

The amount of N for the SeNPs-Chitosan sample appeared abnormally low but increased significantly after etching. Etching also caused the C and F intensity to drop and increased the intensity of N, Na, Se, and P (Table 1).

Figure 3 shows the Se_3D_ core-level scans for both samples before etching. The peak shape for the SeNPs-BSA clearly resolves the 3D spin orbit splitting, characteristic of Se^0^. The peak could be fitted with a doublet with an intensity ratio for Se_3D_^3/2^/Se_3D_^5/2^ of 0.735, an energy offset of 0.85 eV, and an absolute energy position of 54.6 eV, in good agreement with the literature on Se^0^. An additional peak at 53.5 eV is attributed to a reduced form of Se, either Se^I−^ or Se^II−^ or a mix of both, since the full width at half maximum (FWHM) is quite broad. The peak shape for the chitosan sample is significantly different than for the BSA sample and shows a clear oxide component, Se^IV+^ at 58.7 eV, as well as a reduced component at 53.45 eV, indicating that Se^IV+^ was not totally reduced to elemental selenium during the synthesis of the SeNPs-Chitosan nanoparticles. The central peak was fitted with the same Se^0^ doublet as for the BSA sample but with an extra component on the high binding energy side, the origin of which is speculative (labelled as “sub-oxide”). The justification for this component becomes apparent after etching. The FWHM of the doublet is also larger for the chitosan than the BSA, possible because of the less defined chemical state and/or a lower crystallinity or higher roughness. Despite the uncertainty about the precise chemical states of the chitosan sample, Figure 3 clearly shows significant differences in the surface chemistry of the two types of SeNPs.

Following etching, the shape of the Se_3D_ core level for the Chitosan sample appears sharper, with the doublet becoming apparent. The oxide component is also much smaller, and the reduced component larger, as expected from the reducing effect of Ar bombardment. The reduction of the oxide is probably the reason why the metallic component becomes sharper as shown by the smaller FWHM and the better-defined peak shape. The Se_3D_^3/2^/Se_3D_^5/2^ intensity ratio must be 0.735, and therefore, the shape of the doublet is fixed. This is the reason why the component labelled “sub-oxide” must be added in order for the sum to fit the raw data. This was only found for the SeNPs-Chitosan, and this peak for a binding energy of around 56 eV could be attributed to Se^II^ [37]. Interestingly, selenoxide displays peaks around 57 eV [38]; a lower binding energy of selenoxide-containing peptide compounds with a peak at 55.7 eV was observed and assigned as a putative formation of hydrogen bonds between peptides and selenoxide [39]. Still, the origin of our extra component labelled as “sub-oxide” is speculative and would require further in-depth work that was out of the scope of the present study. The etch slightly increased the FWHM of the BSA components, possibly because of surface damage. There was also a clear binding-energy difference for the Se_3D_^5/2^ peak between BSA and chitosan, as shown by the dashed line in Figure 3 and in Table 2. Table 3 provide the parameters of the binding energy and FWHM of the fitting components used in Figure 2. The Se_3D_^5/2^ peak in the BSA sample has a binding energy 0.3 eV lower than the chitosan sample, relative to the C-C bond at 284.8 eV. All other elements (N, C, O, Na, and P) have the same energy.

#### 3.1.3. X-ray Absorption Near-Edge Spectroscopy (XANES) Measurements of the Manufactured SeNPs-BSA or -Chitosan

The internal structures of the SeNPs were highly distorted. The X-ray diffraction pattern was characteristic of an amorphous structure and very little structural information could then be obtained. XANES measurements were then performed on the nanoparticles, XANES being sensitive to the local order around the probed element. This informed on the oxidation state of the probed element, like XPS, but measurements were performed at the Se K-edge, using hard X-rays, that provided information not limited to the surface of the SeNPs.

We first tested our measurement and calculation procedures by measuring and simulating the XANES of both the Se^0^ red and Se^0^ grey allotropes. Initial structures for the calculations were taken from the literature (Appendix A). The optimum adjustment is shown in Figure 4 while the influence of the cluster size (radius R) on the calculated spectra is shown in Figure 5. We managed to reproduce the main features of the Se^0^ experimental XANES spectra with the calculations in term of the positions of the transitions and bumps after the main line (the so-called white line). The optimum radius for the cluster size was found to be R = 5 Å in both cases. A higher radius does not induce further significant changes for the monoclinic calculation, while it decreases the quality of the adjustment for the trigonal calculation when compared to the Se^0^ grey experimental spectrum. Indeed, the rigid structure used for the calculation does not take into account the distribution of distances and angles between the Se-6-chains and Se-8-rings or the possible conformations inside the sample like in amorphous red selenium [38] Another point is that for all the possible structures, when R decreased, the white-line intensity decreased, and the features around 12,665–12,675 eV were less marked (Figure 5).

During the second step, we performed XANES measurements on the nanoparticles as manufactured and after incubation. Selenium was potentially present in the SeNPs in the Se^0^ form (the core of the NP) and selenite (as a trace of the synthesis process). We performed a linear combination fitting fit on the SeNP experimental XANES using calculations as a standard for the Se^0^ spectrum, with the different radius values for the calculation (R values from 3 to 6 Å) to take into account the limited size of the nanoparticle core, and the selenite solution experimental reference XANES spectra obtained in the same experimental conditions (1% selenite aqueous solution sample measured at 10K). The results are shown in Figure 6 for the four samples, with the two nanoparticles as obtained from the manufacturer and the nanoparticles incubated for 24 h. The best fits, as selected using the R-factor values characteristic of the adjustment quality, were always obtained with the “monoclinic − R = 5 Å” calculated spectra (Appendix A). The cores of the as-obtained nanoparticles had the Se^0^ red structure, with a local order similar to the macroscopic Se^0^ red sample. Moreover, the core structure was not affected by the incubation. The distribution between the Se^0^ and the selenite forms inside the samples as determined using the LCF is given in Figure 6B.

Overall, despite the commercial nature of the SeNPs, we have shown that an in-depth characterization of their structure, and particularly their speciation, is of importance. This demonstrates that Se^0^-based SeNPs have the expected structure and stable core of Se^0^, but a small fraction of Se^IV^ is still associated and could also participate in the biological effects prior to or after cellular internalization and in their biotransformation and intracellular fate.

### 3.2. Cytotoxicity Assay and Cell Migration of PC-3 Cancer Cells Exposed to Selenium Nanoparticles

The cytotoxicity was assessed on PC-3 cell monolayers in the presence of increasing concentrations of SeNPs-BSA, SeNPs-Chitosan (0.39–75 μg/mL), or sodium selenite (0.1 to 10 µg/mL) over 24 h. The cell viability measured using MTT assay showed a greater cytotoxicity of sodium selenite treatment when compared to treatment with SeNP-BSA and SeNP-Chitosan, with the later reaching a significant reduction in cell viability for a concentration of 25 µg/mL while for SeNPs-BSA cells, a 47% viability remained at concentrations as high as 75 µg/mL (Figure 7A,B). This dose-response curve allowed to calculate the IC20 at 24 h for SeNP-BSA (1.66 µg/mL), SeNP-Chitosan (0.87 µg/mL), and sodium selenite (1.42 μg/mL). We decided to study the effects of SeNPs at an IC20 concentration that we consider as sub-toxic. The IC20 for sodium selenite could also be derived from Figure 7B and was found to be around 1.1 µg/mL, which corresponds roughly to 6.4 µM, a value that appears to be in line with values from previously reported cytotoxicity tests [40].

We then used a wound healing assay to determine the ability of cells to migrate and repair a wound in the presence of the different selenium compounds. Following the scratching of the cell monolayer, cells were placed in the IncuCyte Live Cell workstation and wound repairing was recorded for 24 h at intervals of 1 h. The calculated wound recovery density (RWD) of PC-3 cells was decreased following treatment with SeNPs-BSA compared to control (*p* < 0.001). Similarly, cells exposed to sodium selenite (Se^IV^) were unable to repair the wound (Figure 7C,D, *p* < 0.001, Appendix A), while the RWD was slightly decreased with SeNPs-Chitosan treatment but found to not be significantly different from control. These results indicated that sub-toxic concentrations of SeNPs-BSA and sodium selenite inhibited cell migration.

### 3.3. Effects of SeNPs-BSA or -Chitosan on Prostate PC-3 Cancer Cell Roughness and Biomechanics

The above results led us to study the PC-3 cell mechanical change, if any, upon treatment with SeNPs. Indeed, evidence has been reported that cancer cells are softer than normal cells and that cellular rigidity decreases with the metastatic process [41]. AFM measurements allowed to obtain information on cell roughness that was calculated with eight cells with 25 µm^2^ area measurements and 1 µm^2^ analysis squares in all groups. Images of typical areas are displayed in Figure 8A–C. The PC-3 control cell-surface roughness (R_RMS_ = 46.6 ± 12 nm) was significantly higher than that observed in selenite-treated cells (R_RMS_ = 30.04 ± 8.5 nm). SeNPs-BSA treatment did not seem to affect surface roughness in comparison with the control (R_RMS_ = 45.16 ± 17.12 nm, *p* > 0.05) and SeNPs-Chitosan significantly decreased PC-3 cell surface roughness in comparison with control (R_RMS_ = 39.3 ± 17 nm). Figure 8D,E). An AFM probe with a colloidal tip was used as a nanoindentor to measure cell stiffness and monitor changes in cell elasticity after 24 h treatment with selenium compounds. Using Hertz mechanics, the elasticity can be calculated from the observed changes in the contact regime of the force curve. Total cell elasticity values are drawn in the frequency curve for control and treated cells. Significant alterations in median values were detected between control and cells exposed to SeNPs-Chitosan and selenite while no significant changes were observed for cells incubated with SeNPs-BSA (Figure 8). These results indicate that the effects on PC-3 stiffness vary between selenium compounds tested and do not necessarily correlate with our results above on PC-3 cell migration. As highlighted by Luo et al., if the Young’s modulus can be used for the characterization of the metastatic propensity of cancer cells, this may not be fully valid when considering the addition of a pharmacological treatment [42] in the form of SeNPs as proposed herein.

### 3.4. HPLC-ICP-QQQ-MS and HERFD-XAS Bulk Speciation of PC-3 Cells Exposed to Selenium Compounds

To investigate the effects of biotransformation within the cells of the Se^0^ core of SeNPs when incubated with PC-3 cells for 24 h at a sub-toxic concentration, the bulk speciation from frozen cell pellets was studied using synchrotron high-energy-resolution fluorescence-detected X-ray absorption spectroscopy (HERFD-XAS). We performed a linear combination fitting of model Se compounds to the HERFD-XAS spectra; the models were selected from the model compound library [43] following PCA and target analyses. The spectra showed that within the cells, main-species Se^0^ remained with the presence of selenodiglutathione GS-Se-SG, the latter being in a higher proportion for PC-3 cells treated with SeNPs-Chitosan compared to SeNPs-BSA. Selenocystamine for alkyl diselenide (RSeSeR) species was also found necessary for spectral fitting. Of note, no Se^0^ was found in cells exposed to sodium selenite and this supports that no elemental selenium was formed during the metabolization of selenite in our conditions (sub-toxic concentration of sodium selenite). The total selenium content calculated from the cell pellets (Figure 9C) ranged from 8.0 ± 1.7 ppb (2sd) for control to 69.1 ± 29.3 (2sd) ppb for cells exposed to SeNPs-Chitosan (eight-fold increase), 430.7 ± 129.9 (2sd) ppb with sodium selenite (fifty-three-fold higher than control), and 501.2 ± 109.7 (2sd) ppb with SeNPs-BSA (sixty-two-fold higher than control). This allows to normalize the % of Se species detected in a cell culture supernatant after 24 h incubation for the different conditions tested (Selenite, SeNPs-BSA, and SeNPs-Chitosan treatment). A two-factor increase in selenite or the number of Se_IV_ species was observed in the media for all conditions when compared to control. Higher percentages of Se species, thiols and selenate, were also detected in the supernatant when compared to control (Figure 9D). Overall, these results show the SeNP metabolization by cells leads to the formation of Se species similar to that by cells exposed to aqueous selenite with mainly increased GS-Se-SG and diselenide species. This suggests an oxidative stress response inside cells as already observed in ovarian cancer cells [44]. Indeed, sodium selenite is reduced to selenodiglutathione and then to hydrogen selenide by glutathione. Previous work has documented the presence of diselenide compounds in cancer cells when incubated with sodium selenite [45] or with methylselenocysteine [46], and the authors suggest that this presence is indicative of change in the intracellular redox environment with a possible concomitant loss of function of selenoproteins due to the formation of diselenide bonds, thus being detrimental to cell viability [45]. Of importance, diselenide selenocystine (CysSeSeCys), which is the oxidized form of selenocysteine, has been shown to indicate an increase in thioredoxin reductase activity [47], but also to induce apoptosis in certain cancer cells through increased intracellular reactive oxygen species production [48].

### 3.5. Intracellular Distribution of SeNPs in PC-3 Cells

The X-ray-fluorescence elemental mapping of plunge-frozen and freeze-dried cells allowed us to analyze cells at room temperature at the Diamond I14 nanoprobe beamline. A compromise had to be made between the scanning time and number of cells for statistical relevance, and a 200 nm step size was used for XRF mapping (Appendix A). The average (value ± 2 SDs) cellular concentrations of Se in µg/g dry weight could be calculated and were found to be 111 ± 35 µg/g (N = 7), 34 ± 12 µg/g (N = 6), and 8 ± 5 µg/g (N = 4) after the 24 h treatment with SeNPs-BSA, SeNPs-Chitosan, and sodium selenite, respectively, while no Se could be detected in control due to a concentration below the limit of detection. The potassium (K) distribution was found to be 10,930 ± 22,070 µg/g (N = 7) for control cells, 91,800 ± 30,340 µg/g (N = 7) after SeNPs-BSA treatment, 82,767 ± 27,122 µg/g (N = 6) for SeNPs-Chitosan treatment, and 71,520 ± 9267 µg/g (N = 4) for sodium selenite treatment. These concentrations of K values are in the range of values for well-preserved cells after the freeze-drying procedure [49].

To explore further the Se intracellular distribution, we decided to study cells in their near-native state i.e., frozen–hydrated using a quite-unique cryo-X-ray-fluorescence nanoprobe that provides an X-ray spot size down to 20 nm and sensitivity down to a few hundred to a few thousand atoms (Figure 10). We could have used a 50 nm beam, but the throughput was low due to the high resolution and concomitant necessary long time for analysis. This analysis was mainly devoted to imaging the Se cellular localization, but the Se average elemental areal concentration could be derived. The average (value ± 2 SDs) Zn areal mass (ng/mm^2^) values were all in a similar range (from 0.166 for SeNPs-Chitosan to 0.261 for control) while for Se, the expected very low concentration in control cells could be detected (0.003 ± 6 × 10^−4^ ng/mm^2^, N = 3), and upon SeNP treatment (IC20 for 24 h), the highest cellular Se content was for SeNPs-BSA (0.191 ± 0.09 ng/mm^2^, N = 5), followed by the positive control with sodium selenite (0.066 ± 0.015 ng/mm^2^) and finally SeNPs-Chitosan (0.017 ± 0.04 ng/mm^2^). This trend was in agreement with the total Se content obtained from bulk PC-3 cell pellets using HPLC-QQQ-ICP-MS (Figure 9C). The cellular distribution was mainly punctual within the cytosol with a hotspot of high intensity that we could assign, with confidence, to small SeNP aggregates, while for control, Se distribution was fuzzy and very low, and after selenite treatment, Se was found also spread in the cytosol with very rare Se hot spots, an observation already reported using synchrotron XRF for lung cancer cells exposed to a high concentration (corresponding to ~IC50) of sodium selenite [46].

We had the possibility to test, on PC-3 cells treated with SeNPs-BSA, a recent experimental facility, i.e., cryo-optical correlative microscopy. The same frozen–hydrated cells could be imaged in optical fluorescence with a prior labeling of organelles—herein, Lysosomes using a live cell fluorescent probe Lysosomes-RFP CellLight™, BacMam 2.0 ((Thermo Fisher Scientific, Waltham, MA, USA), overnight incubation prior to the 24 h SeNPs treatment, followed by the plunge-freezing procedure). The very same cells could be imaged using a cryo-XRF nanoprobe (Figure 11 and Appendix A). These preliminary data support, at least in the case of the SeNPs-Chitosan treatment, that hotspots of Se in the cell correlate with the positions of the lysosomes. This was expected, as it is widely accepted that endocytosis is the main entry mechanism for nanoparticles, followed by accumulating in acidic vesicles (early endosomes) and mostly ending in lysosomes through the endo/lysosomal pathway [34,50].

To gain further insight into the localization of SeNPs in the cell interior, the TEM imaging of thin sections of high-pressure-freezing PC-3 cell pellets was obtained to access ultrastructural information. It can be observed that organelles in control (PC-3) are very well preserved and cells display a granular endoplasmic reticulum with ribosomes and healthy mitochondria. After incubation with a sub-toxic concentration of SeNPs, some nanoparticle clusters appear in close contact or inside mitochondria, particularly in SeNPs-BSA treatment with a high content of SeNPs clusters, while SeNPs appear more in intracellular vesicles such as endosomal/lysosomal compartments (Figure 12B,C). In the case of selenite treatment, even at a sub-toxic concentration, it can be observed that there are some alteration of organelles and some vacuoles, possibly autophagic vacuoles. These observations for SeNPs are in line with XRF results, with a higher content of Se in PC-3 cells treated with SeNPS-BSA and a lower content for SeNPs-Chitosan with some colocalization with lysosomes as observed above. The intracellular presence at a higher content of SeNPs-BSA could be the result of improved uptake and targeting through the albumin receptor-mediated endocytic pathway [51]. Localizations of some clusters of nanoparticles inside mitochondria have already been reported [52,53], and additional studies are necessary in order to decipher the intracellular transport pathway of some SeNPs into mitochondria.

Overall, the results show some differences between the exposure of PC-3 cells to SeNPs-BSA and to SeNPs-Chitosan. The cellular uptake of SeNP-BSA is higher than that of SeNP-Chitosan; however, the IC20 of SeNP-BSA is higher than the IC20 of SeNP-Chitosan. Generally, there is a clear influence of many factors such as the cell type or route and duration of exposure, concomitantly to the shape agglomeration (or not) and surface coating of nanoparticles. A possible explanation for the above difference in uptake could be that SeNPs-Chitosan solution appears much less stable, with a tendency to aggregate, leading to larger particle sizes that are far from optimal for efficient intracellular uptake, contrary to SeNPs-BSA. Despite the much higher uptake of SeNPs-BSA, the higher IC20 could be explained by the BSA coating that has been reported to reduce the cytotoxicity of a number of types of nanoparticles, possibly preventing interactions between the reactive surface of the nanoparticles with the plasma membrane and mitigating the generation of reactive oxygen species [54].

Of note, some immunofluorescence experiments have shown that upon the incubation of SeNPs at sub-toxic concentrations (in our case, below ~10 µM), the percentages of nuclear γH2AX foci and highly damaged cells defined by Viau et al. (nuclei showing more than 15 γH2AX foci per cell) [55] and micronuclei considered to be the cytogenetic reflections of the unrepaired DSB propagated to the mitotic phase were found to be similar to those in control cells. Taking advantage of well-documented double-strand break (DSB) induction upon controlled X-ray irradiation (2 Gy), the impact of the post-irradiation of sub-toxic doses of SeNPs on the repair kinetics of the cell lines did show a delay when SeNPs-BSA was used compared to SeNPs-Chitosan (Appendix A). According to these preliminary results and although speculative, it would be of interest to perform further studies to explore whether the aforementioned impact could be linked to some delay in the nuclear shuttling of the ATM protein and slowing down of the migration capabilities of PC-3 cells.

## 4. Conclusions

It has been observed that selenium (Se) can inhibit cancer cell growth, but the known toxicity associated with its soluble form limits its use as an anticancer agent. The use of functionalized inorganic Se^0^-based nanoparticles has the potential to overcome this limitation and can be administered at far higher concentrations, being effective in vitro and in vivo.

The present work has provided a study of the impact and fate of functionalized SeNPs on prostate cancer cells (PC-3). We have shown the importance of an in-depth characterization of the speciation of synthesized Se^0^ nanoparticles as the presence of a residual or oxidized form of Se could influence the in vitro experimental results. Even at the sub-toxic dose used herein, the SeNPs-BSA was found to slow down the migration of invasive prostate cancer, but did not seem to be associated with a change in the nanomechanical properties of the cell cytoskeleton. This intracellular fate of these nanoparticles in aggressive prostate cancer cells was investigated using synchrotron X-ray spectroscopic techniques. Using an inhibition concentration resulting in an 80% cell viability that was effective in reducing cancer cell proliferation in vitro, we demonstrated that the nanoparticle intracellular concentration was quite low and mostly found as clusters in lysosome-related organelles or mitochondria. Still, some selenium was also found within the cytosol of the cells. Despite a low intracellular concentration, the high-energy-resolution fluorescence-detected X-ray absorption near-edge structure allowed us to show that the biotransformation/degradation of nanoparticles into Se species is different from that of those nanoparticles produced under exposure to a soluble form of selenium, i.e., sodium selenite. We could evidence the presence in cells of redox-active selenium compounds (selenite, selenodiglutatione, and the diselenide type of the compound RSeSeR) that generate selenide under reduction but also can generate superoxide in the presence of thiols and oxygen, resulting in the formation of various ROS. The differences observed in the uptake of Se and localization between SeNPs-BSA (more in mitochondria) and SeNPs-Chitosan (lysosomes) provide a basis to explain the different mechanism of toxicity that is also different from that of the aqueous solution of sodium selenite used as positive control. Still, the contribution to some cytotoxicity effects of selenite formed during incubation within the cell culture media cannot be ruled out. The presence of SeNPs-BSA aggregates in mitochondria may have importance in the mechanism of the action of the SeNPs to slow down the proliferation of prostate cancer cells. This seemed to have been supported by the preliminary genotoxicity experiments. The impact of SeNPs-BSA appeared to reduce the double-strand break DNA repair in PC-3 cells, which was correlated to the migration regression. Our findings highlighted new insights into various mechanisms of cytotoxicity of SeNPs and opened some very interesting perspectives in genotoxicity investigations.

## Figures and Tables

**Figure 1 nanomaterials-13-02999-f001:**
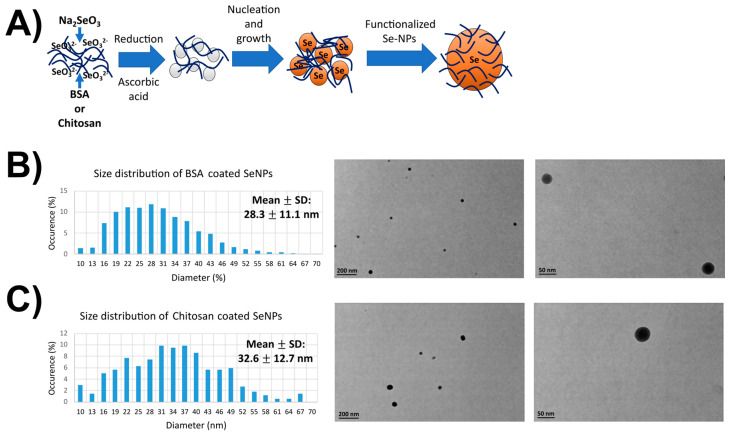
TEM images and the size distribution of SeNPs. (**A**) Scheme of the SeNP synthesis and functionalization. (**B**) Size distribution and typical TEM micrograph of the BSA-coated SeNPs. (**C**) Size distribution and typical TEM micrograph of the chitosan-coated SeNPs. Over 100 micrographs for each condition were processed to derived the size diagram above. TEM images have been taken at ×23,000 and ×130,000 magnifications.

**Figure 2 nanomaterials-13-02999-f002:**
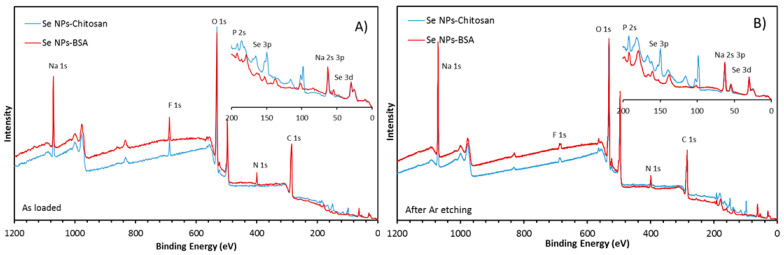
Survey scans of the as-prepared SeNPs-Chitosan and SeNPs-BSA. The inset shows the 0 eV to 200 eV region in more detail: (**A**) as loaded, (**B**) after Ar cluster etching. The core levels of interest have been labelled.

**Figure 3 nanomaterials-13-02999-f003:**
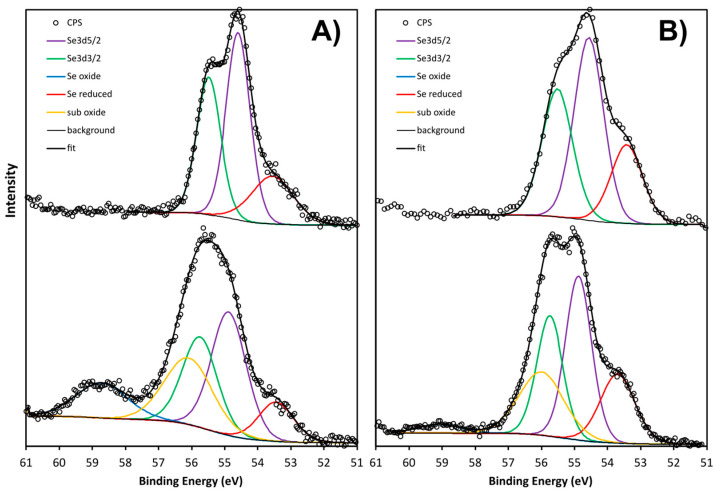
XPS scan of the Se 3D core levels of the SeNPs-Chitosan and SeNPs-BSA (**A**) before Ar etching and (**B**) after Ar etching, showing the different chemical states.

**Figure 4 nanomaterials-13-02999-f004:**
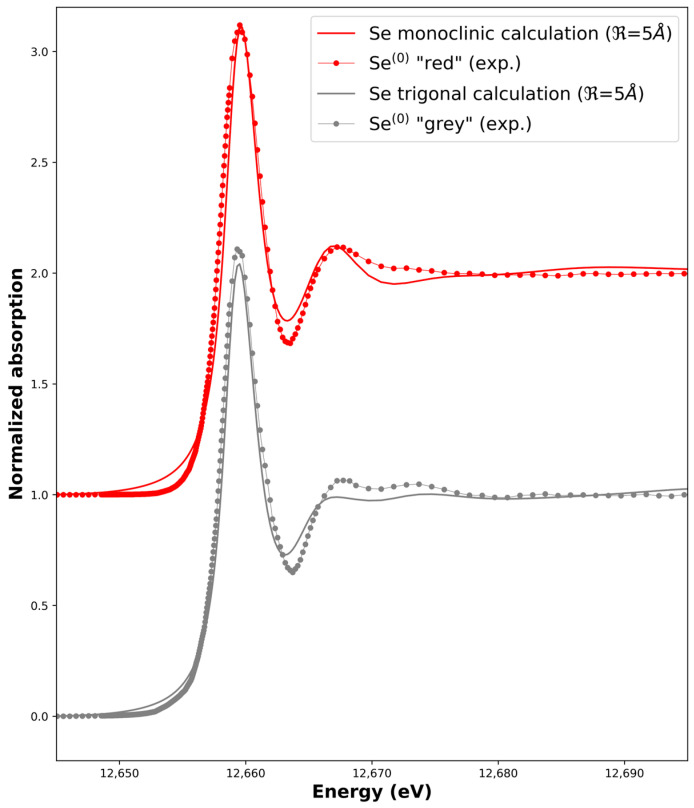
Calculated X-ray Absorption Near-Edge Spectra of the monoclinic and trigonal structures (R = 5 Å) compared to experimental ones, i.e., Se^0^ red and Se^0^ grey experimental spectra, respectively. Data have been shifted for clarity.

**Figure 5 nanomaterials-13-02999-f005:**
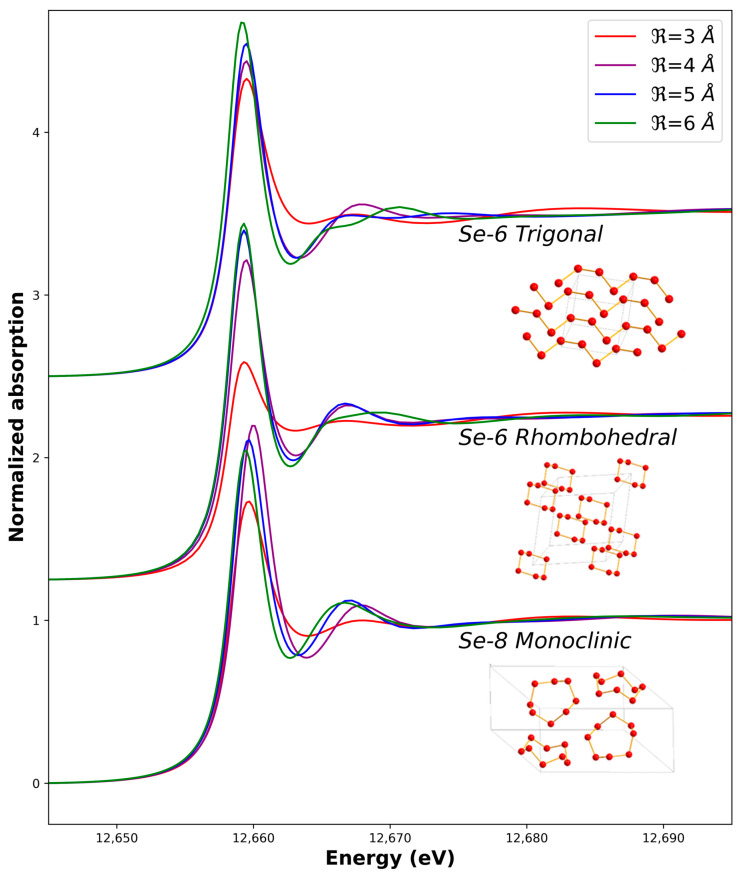
Calculated X-ray Absorption Near-Edge Spectra based on different structures and cluster sizes (see Appendix A for structural details).

**Figure 6 nanomaterials-13-02999-f006:**
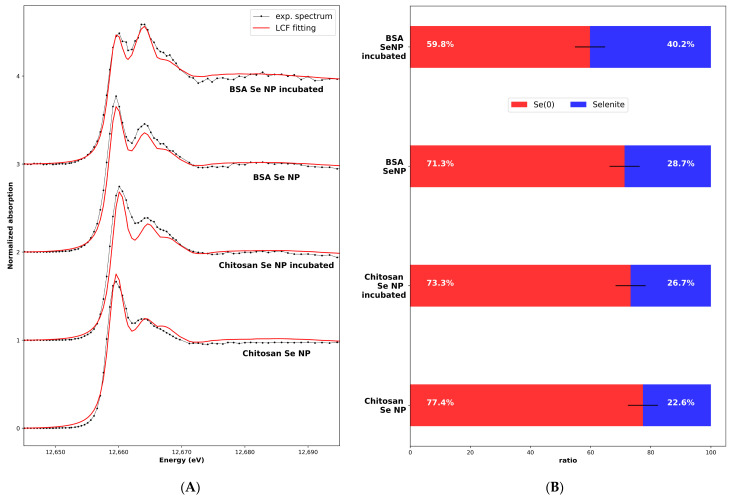
(**A**) Experimental and least-square-fitting-calculated spectra of SeNPs-BSA or -Chitosan, as-obtained and incubated for 24 h. The calculated Se^0^ XANES spectrum (corresponding to the monoclinic structure and a cluster size of R=5 Å) and experimental selenite XANES spectrum (sample: 1% selenite aqueous solution) were used for the LCF fitting. (**B**) Se^0^ and selenite ratio, estimated from the LCF. Error bar: 5%.

**Figure 7 nanomaterials-13-02999-f007:**
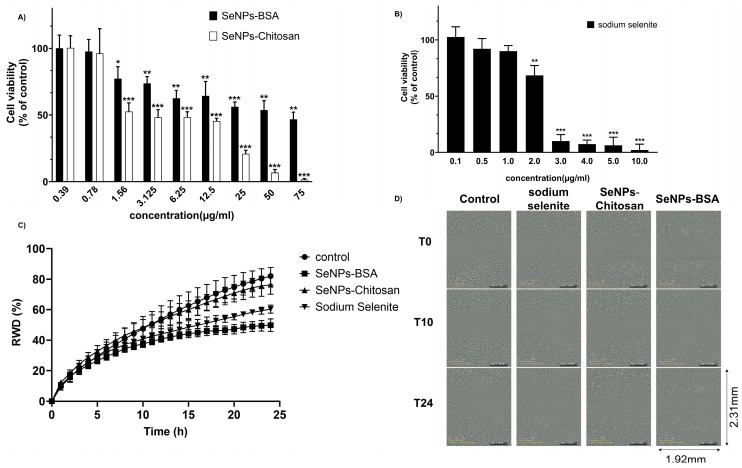
Effect of selenium compounds on PC-3 cells: (**A**) Effect of SeNPs-BSA and SeNPs-Chitosan on cell viability; (**B**) effect of sodium selenite on cell viability. Cells were cultured in the presence of increasing concentrations of each selenium compound. After 24 h, MTT assay was performed to measure cellular viability. Values are expressed in percentages as the means ± SDs of at least three separate experiments. *—statistically significant vs. control at *p* < 0.05. **—statistically significant vs. control at *p* < 0.01. ***—statistically significant vs. control at *p* < 0.001. (**C**) Wound healing ability of PC-3 cells treated with different IC20 concentrations of selenium nanoparticles (SeNPs-BSA and SeNPs-Chitosan) and sodium selenite using the IncuCyte live cell workstation and wound scratch assay. This presents the average relative wound density (RWD in %) from *n* = 5 identically prepared experimental replicates, and the error bars in (**C**) indicate one standard deviation from the mean. Representative wound healing assays using (**D**) PC3-cells at time 0, 10, and 24 h are shown. Selenium compounds (0 µg/mL) were used as the controls. Magnification: ×10; see Appendix A for details.

**Figure 8 nanomaterials-13-02999-f008:**
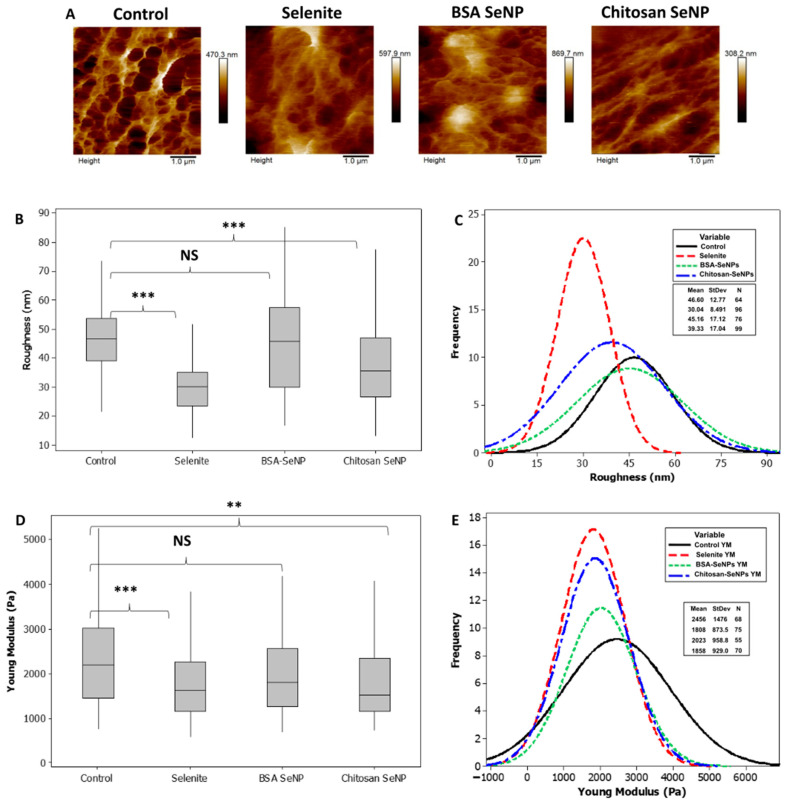
SeNP effect on prostate cancer cells’ (PC-3) close topography, surface roughness, and stiffness. Statistical significance was determined using the Mann–Whitney test with the following used symbols: NS = *p* > 0.05, ** *p* < 0.01, and *** *p* < 0.001.

**Figure 9 nanomaterials-13-02999-f009:**
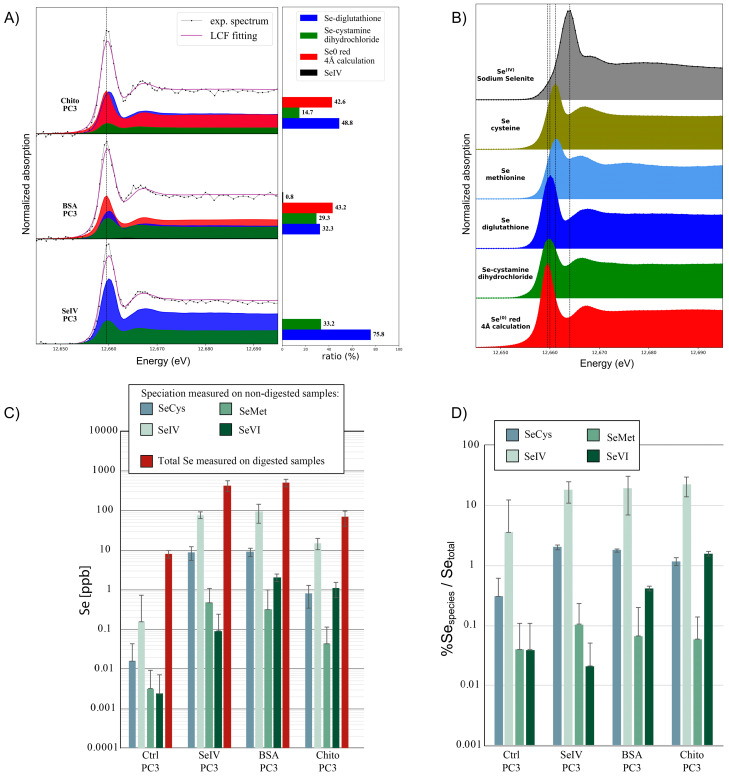
Evaluation of selenium speciation using synchrotron high-energy-resolution fluorescence-detected X-ray absorption spectroscopy (HERFD-XAS) on PC-3 cells after 24 h treatment of SeNPs and sodium selenite treated at IC20. (**A**) Linear combination fitting (LCF) of samples by reference selenium species. (**B**) HERFD-XAS spectra for some reference Se compounds (in frozen-solution state unless otherwise specified). From top to bottom: sodium selenite (Se^IV^), S-methyl seleno L-cysteine, L-Selenomethionine, selenodiglutathione, Se-cystamine dihydrochloride (solid), red elemental selenium (Se^0^, solid). A dashed line is used as a guide for the position of the maximum of the white line of the different references. Selenium 0 (red) was calculated to obtain a reference spectrum that was used in the fits of all samples. Measurements were made in duplicate on two different samples. (**C**) Evaluation of different Se species in PC-3 cell supernatants and total Se concentration in PC-3 cell pellets using HPLC-QQQ-ICP-MS after 24 h treatment with SeNPs or sodium selenite. (**D**) Percentage of Se-chemical species normalized by the total Se quantity found in cell pellets. SeCys: Selenocysteine, SeMet: Selenomethionine; Se^IV^ and Se^VI^ oxidation states can be associated with selenite and selenate compounds.

**Figure 10 nanomaterials-13-02999-f010:**
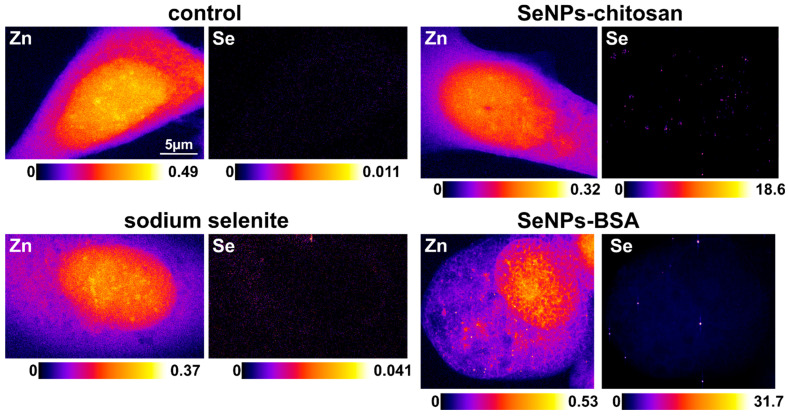
Cryo-X-ray-fluorescence nanoimaging elemental distribution maps of PC-3 cells from control (no treatment with Se compounds) and PC-3 cells treated for 24 h at IC20 sub-toxic concentration of SeNPs-BSA, SeNPs-Chitosan, and sodium selenite compounds. The range of elemental areal mass (quantified from standards and expressed in ng/mm^2^) is given in the bottom of each map.

**Figure 11 nanomaterials-13-02999-f011:**
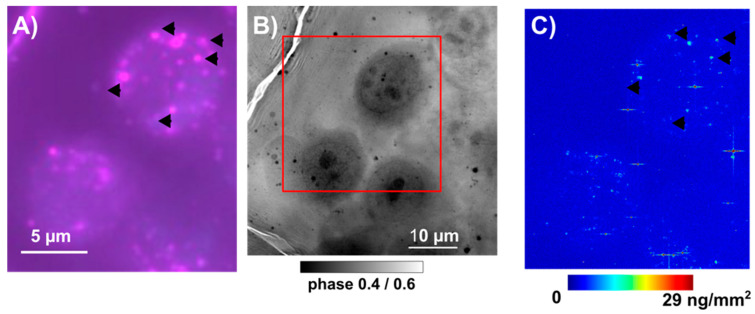
Cryo-correlative imaging of PC-3 cells exposed to IC20 concentration of SeNPs-Chitosan for 24 h. (**A**) Cryo-optical fluorescence imaging (120K) of vitrified PC-3 cells cultured on Si_3_N_4_ membranes. Live cells were first incubated overnight with Lysosomes-RFP CellLight™, BacMam 2.0 reagent following manufacturer recommendations, then cells were exposed for 24 h to SeNPs-Chitosan at IC20 concentration followed by plunge-freezing procedure. (**B**) Hard X-ray phase contrast imaging of the same cells, at the ID16A cryo-nanoimaging allow to show some structural details, mainly the nuclear region and shape of each cell (the red square show the corresponding region imaged in cryo-optical fluorescence in (**A**) and the XRF scanning region in (**C**)). (**C**) Cryo-XRF nanoimaging at 50 nm step size of the selenium (Se) distribution in PC-3 cells exposed to SeNPs-BSA. This allows to correlate some hotspots of Se (arrow heads) with the positions of lysosomes (arrow heads in (**A**)).

**Figure 12 nanomaterials-13-02999-f012:**
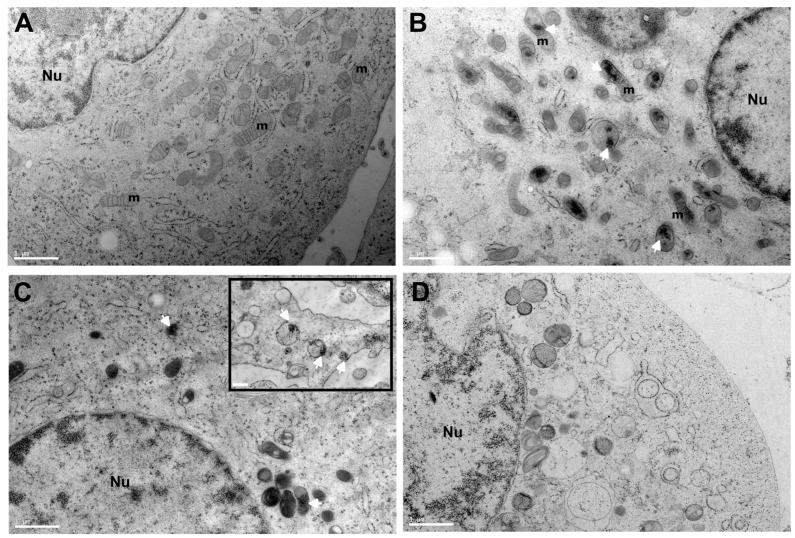
Transmission electron microscopy (TEM) analysis of PC-3 cells treated with SeNPs. (**A**) Electron micrograph of PC-3 control cells (incubation without Se compounds). PC-3 cells after 24 h incubation of sub-toxic concentration of SeNPs-BSA (**B**), SeNPs-Chitosan with the insert showing a higher magnification of another region of the cytosolic part of the cells (**C**), and sodium selenite (**D**). A magnification of ×2900 and a scale bar of 1 µm for all images, and a magnification of ×4800 for the insert with a scale bar of 0.5 µm, have been used. Nu: cell nucleus, m: mitochondria. Cluster of nanoparticles: indicated by the white arrow.

**Table 1 nanomaterials-13-02999-t001:** The surface compositions of the main constituents by atomic % from the survey scans of Figure 2.

%	C	N	O	F	Na	Se	P
SeNPs-Chitosan	50.2	0.4	38.5	3.1	6.3	0.8	0.6
SeNPs-Chitosan Ar etched	39.6	1.7	41.8	1.7	10.6	1.6	3.1
SeNPs-BSA	51.2	3.6	33.3	4.3	5.9	0.3	0.7
SeNPs-BSA Ar etched	46.7	4.6	33.6	2.2	10.1	0.8	2

**Table 2 nanomaterials-13-02999-t002:** Summary of the contributions of the different chemical states for the Se 3D peak shape as obtained from the curve fitting shown in Figure 3 and Figure 4. Se^0^ is the sum of the intensities of both Se_3D_^3/2^ and Se_3D_^5/2^ components.

%	Se^0^	Se Oxide	Se Reduced	“Se Sub-Oxide”
SeNPs-Chitosan	53.9	13	10	23.1
SeNPs-Chitosan Ar etched	56.9	2.9	18.4	21.8
SeNPs-BSA	79.4	0	20.6	0
SeNPs-BSA Ar etched	78.7	0	21.3	0

**Table 3 nanomaterials-13-02999-t003:** Binding energy and full width at half maximum (FWHM) of the fitting components used in Figure 3. The binding energies have been calibrated to the C-C line at 284.8 eV.

	Se_3D_^5/2^	Se_3D_^3/2^	Se^IV^	Se^−II^	“Sub-Oxide”
Binding energy (eV)					
SeNPs-Chitosan	54.88	55.75	58.73	53.45	56.10
SeNPs-BSA	54.59	55.48	-	53.55	-
SeNPs-Chitosan Ar etched	54.87	55.74	59.07	53.70	55.99
SeNPs-BSA Ar etched	54.56	55.51		53.42	
FWHM (eV)					
SeNPs-Chitosan	1.24	1.24	1.84	1.26	1.7
SeNPs-BSA	0.85	0.85	-	1.5	-
SeNPs-Chitosan Ar etched	0.92	0.92	1.5	1.23	1.59
SeNPs-BSA Ar etched	1.04	1.09	-	1.14	-

## Data Availability

Data will be made available upon reasonable request. Se HERFD-XANES spectra are available at https://doi.org/10.26302/SSHADE/EXPERIMENT_CB_20190408_001 (accessed on 1 April 2020).

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
