# Peer review of "Intracellular Fate of Sub-Toxic Concentration of Functionalized Selenium Nanoparticles in Aggressive Prostate Cancer Cells"

_nanomaterials, 2023, doi:10.3390/nano13232999_

Round 1
Reviewer 1 Report
Comments and Suggestions for Authors
In this study, the authors report intracellular localization and fate of sub-toxic concentrations of SeNPs capped with BSA and Chitosan in prostate cancer PC-3 cells by various spatially resolved and speciation techniques. SeNPs displayed different intracellular uptake with SeNPs-BSA preferentially accumulated into mitochondria. In addition, SeNPs-BSA slow down the migration of PC-3 cells. Understanding the interaction of SeNP and biological systems is crucial for its anti-cancer application. Before it is accepted for publication, there are some comments should be addressed.
1. Introduction should be revised to highlight the importance of the key scientific issue studied in this paper. In addition, before introducing BSA and Chitosan(L63), it is suggested to explain why functionalization of SeNP is necessary.
2. L350-353. Besides TEM, the hydrodynamic size and PDI of NPs in water and cell culture medium should be provided to support the statements.
2) The data in Figure 2 and Table 1 do not support that BSA and Chitosan have been coated on SeNPs successfully. In addition, what’s the mass ratio of SeNP in the functionalized SeNP.
3) Please check the data of SeNP-Chitosan and sodium selecnite in Figure 7C, which are conflict to the data in Figure 7D and Figure S2.
4) Cell uptake of SeNP-BSA is higher than that of SeNP-Chitosan; however, the IC20 of SeNP-BSA is higher than that of SeNP-Chitosan. Please discuss the relationship of cell uptake, selenium speciation, and toxicity of the SeNPs.
5) A lysosome fluorescent probe was used to prove that SeNPs are in lysosomes. But, the conclusions strength that SeNPs-BSA preferentially accumulated into mitochondria. Why not used a mitochondrion fluorescent probe to prove it, as it is difficult to identify SeNPs using TEM.
6) Please keep the space between numbers and units.
Author Response
We thank the reviewer for her/his positive comments and helping in improving the manuscript, below is our reply to comments:
1-Following your recommendation, we have revised the introduction to highlight main findings of the work
a sentence has been added at L60 with also a recent reference (Hu, R., Wang, X., Han, L. and Lu, X., 2023. The Developments of Surface-Functionalized Selenium Nanoparticles and Their Applications in Brain Diseases Therapy. Biomimetics, 8(2), p.259.)
2- We thank the reviewer for is useful remark, we have measured the PDI of nanoparticles and for SeNPs-BSA PDI is of 0.123 ± 0.002 which was considered monodisperse. For SeNPs-chitosan an average size of 320 ± 221 nm and PDI of 0.220 ± 0.012 and was considered as polydisperse (referrer to Clayton KN, et al.. Physical characterization of nanoparticle size and surface modification using particle scattering diffusometry. Biomicrofluidics 2016;10, https://doi.org/10.1063/1.4962992.). We did effort to measure in cell culture medium but cells are growing in medium with 10% SVF, the protein content ect... spoil our measurements and we could not determine the PDI and size of nanoparticles in culture medium.
The following paragraph has been modified as below:
BSA-coated Se-NPs (32.6±12.7 nm) present a spherical and homogeneous form contrary to chitosan-coated SeNP (28.3±11.1 nm) which are less homogeneously spherical and which tend to form aggregates with increasing concentrations (Figure 1). Indeed, some Dynamic Light Scattering and Zeta Potential measurement were conducted using a ZetaSizer instrument (Malvern Instruments, Malvern, UK). The polydispersity index support above findings with a value of 0.220 ± 0.012 for SeNPs-chitosan which was considered as polydisperse, while for SeNPs-BSA the PDI was found to be 0.123 ± 0.002 which was considered monodisperse. Indeed, SeNP-chitosan had a zeta-potential of 16.4 ± 4.4mV that is within the –30 mV and +30 mV range known to indicate poor stability of the nanoparticles and very likely aggregation or agglomeration while the value of -51.2 ± 15.8 mV for SeNPs-BSA indicate a very high stability of the solution. We noticed that the average hydrodynamic size of SeNPs-BSA was 108 ± 30 nm and 320 ± 221 nm for SeNPs-chitosan this value with the ZetaSizer is higher than the supplier’s specification while the very precise measurements through series of TEM images confirmed those specifications. This can be explained by the fact that the SeNPs-solution is polydisperse and the presence of bigger particles could contribute to shift the measured particles size towards larger values through an increase light scattering.
2) This is an interesting comment raised by the reviewer. In fact, these nanoparticles were commercial and despite efforts we could not get information about % BSA or Chitosan grafting by the company because they coud not deliver the information that is confidential and proprietary from Nanocs, we have the initial solution which is 2mg/ml Se , further as mentioned in the manuscript for example for BSA the disulfide bonds and free thiol groups of BSA are difficult to ascertain and quantify as the Se3s line overlaps with the S2s and Se3p line overlaps with S2p. The surface etching show us that it is different between BSA coated and Chitosan coated but XPS could not differnetiate and be quantitative to % grafting obtained in this commercial product
3) We thank the reviewer for this remark, indeed an error in the labels order for the figure 7D and S7 panels occurs we did correct it, thanks for highlighting it
4)The following paragraph was added as requested by the reviewer's remark:
Overall, the results show some differences between exposure of PC-3 cells to SeNPs-BSA or SeNPs-chitosan. The cellular uptake of SeNP-BSA is higher than that of SeNP-Chitosan; however, the IC20 of SeNP-BSA is higher than the IC20 of SeNP-Chitosan. Generally, there is a clear influence of many factors such as the cell type or route and duration of exposure concomitantly to the shape agglomeration or not and surface coating of nanoparticles. A possible explanation for the above difference in uptake could be that SeNPs-chitosan solution appears much les stable with a tendency to aggregate leading to larger particles size far from optimal for efficient intracellular uptake contrary to SeNPs-BSA. Despite much higher uptake of SeNPs-BSA the higher IC20 could be explained by the BSA coating that was reported to reduce cytotoxicity of number of type of nanoparticles, possibly preventing interactions between the reactive surface of the nanoparticles with the plasma membrane and mitigating the generation of reactive oxygen species (Fröhlich, E., 2012. The role of surface charge in cellular uptake and cytotoxicity of medical nanoparticles. International journal of nanomedicine, pp.5577-5591.)
5) This correlative imaging was really new and not routine (in development) so we did this pilot results on PC-3 exposed to SeNPs-BSA to show the interest to of this CLEM imaging with XRF and for sure require further experiment when becoming a regular tool, still as XRF and cryo-optical fluorescence imaging (not a confocal) are both 2D projection of a volumetric object the cell it is indicative of a correlation but cannot fully ascertain the colocalisation with organelle labelled, a Tomography experiment would be needed, and at the moment TEM is the best technique to have access to interior of the cell with very high resolution at ultrastructure. That is why we did not go further for CLEM between for example mitotracker gree and XRF.
6) We perform the appropriate corrections accordingly
Reviewer 2 Report
Comments and Suggestions for Authors
In this paper, the authors carried out a detailed characterization of the composition of selenium nanoparticles stabilized by BSA and chitosan using modern methods of research, and also described the interaction of nanoparticles with cancer cells. The work is well structured, contains a large amount of data, and is well designed. However, there are a number of remarks to the work that should be taken into account.
1) Authors should uniformly write selenium in the text
2) Authors should correct the abstract keeping only the data directly obtained by them in the paper.
3) All abbreviations should be deciphered the first time they are mentioned.
4) Most of the references are quite long published. Some of them should be updated.
5) It is necessary to standardize the concentrations of working solutions of nanoparticles and sodium selenite normalizing them to selenium. Thus, sodium selenite contains only 45.658 % of selenium, i.e. a solution with a concentration of 2 mg/mL contains only 0.91 mg/mL of selenium. For nanoparticles it is also necessary to adjust the concentrations to the selenium concentration, because the selenium nanoparticles themselves are covered with a polymer layer that contributes to the sample mass.
6) There are no data on the cytotoxicity of BSA and chitosan in experiments testing the cytotoxicity of nanoparticles and sodium selenite, which would allow us to separate the effect of coating on the results obtained.
7) The authors should clarify what they mean by shape and form of nanoparticles?
8) TEM data and their interpretation are questionable: Firstly, the microphotographs of the samples are unsuccessfully chosen to show a very small number of nanoparticles. The manuscript should be provided with overview micrographs of nanoparticles with sufficient number of particles. In addition, the presented micrographs do not correspond at all to the data presented in Figures 1 b, c.
Second: in the text of Section 3.1.1, the authors state that the average sizes of BSA-capped SeNPs and Chitosan-capped SeNPs are 32.6 nm and 28.3 nm, respectively, whereas the diagrams in Figure 1-b and 1-c show different data.
9) The authors should clarify in what form selenite ions are present in the composition of selenium nanoparticles. The authors should also consider the possibility of selenite ions formation during sample preparation.
10) Figures 7-a and 7-b require the addition of a control.
11) The authors should explain the rather contradictory data on the initially higher toxicity of inorganic forms of selenium (sodium selenite), which the authors mention in the introduction referring to the relevant works of other authors; the higher cytotoxicity of chitosan-coated SeNPs and the higher amount of selenium inside the cell for BSA-stabilized SeNPs. It would be good to provide this explanation with the results on determination of zeta potential of nanoparticles and cell surface.
Comments on the Quality of English Language
The article contains a number of minor typos and errors and needs to be checked by a native speaker.
Author Response
1- We use selenium as requested by reviewer
3- We perform the appropriate modification accordingly
4- We updated some references and remove some old or unnecessary references.
5- It is a good point of the reviewer and we did not explain it well. So we did correct the information in material and methods. The commercial solution is provided with a 2mg/ml Se certified by the company Nanocs - so the concentrations used were for Se; the % coating in BSA or Chitosan were not disclosed by the company as they keep it confidential and proprietary of Nanocs. The selenite was made to have also 2mg/ml Se stock solution and all IC20 solution are based on a Se concentration
6- We agree with the reviewer for Chitosan that could slightly contribute to some cytotoxicity eventually but no studies using chitosan-based NPs to our knowledge study in detail such effect, on contrary the BSA is fully biocompatible and not cytotoxic. we added this point about a potent cytotoxicity of chitosan that would require further specific studies using chitosan of different grade or Mw to study its impact on PC-3 cells.
7- the sentence was modified and we essentially focus on a round shape, shape is of importance as a rod-like shape for example or round shape ... have impact also on intracellular uptake of nano-object.
8- The data of the processing of all TEM micrographs (N=100) that results in the diagram of NPs size distribution are at disposal if necessary upon request, the micrograph proposed was mostly chosen to display what images that were processed looks like and zoom to show example of particle with size below 50 nm.
for the other point raised by the reviewer: The average size of the NPs is correct but of course we show the distribution of size of NPs analyzed in large number of EM micrograph to achieve statistical significance and to be transparent on results for the reader. This gaussian distribution is expected with more or less narrow distribution around max peak of the distribution (stable monodisperse solution as for teh SeNPs-BSA), indeed for SeNPs-chitosan solution found to be polydisperse an expected broader distribution of NPs size is found. That is why we conclude that the SeNPs-chitosan are less stable and could form aggregates. details about hydrodynamic size and PDI are reported and paragraph updated accordingly.
9- As mentioned in the XAS results and HPLC-ICP-MS, the form is Se(iv), we add the good remark about possible selenite ions formation during sample preparation/incubation as it cannot be ruled out from our results (L800).
10- The classical data graph representation in biology for such cytotoxicity test are with data expressed as a % of control already, the data for control (cells without any exposure to Se compounds) are from regular 100% viable cultured cells that is always check when cells are sub-cultured.
11- We did improve the manuscript following the remarks of the reviewer and add the 2 following paragraphs in the text
BSA-coated Se-NPs (32.6±12.7 nm) present a spherical and homogeneous form contrary to chitosan-coated SeNP (28.3±11.1 nm) which are less homogeneously spherical and which tend to form aggregates with increasing concentrations (Figure 1). Indeed, some Dynamic Light Scattering and Zeta Potential measurement were conducted using a ZetaSizer instrument (Malvern Instruments, Malvern, UK). The polydispersity index support above findings with a value of 0.220 ± 0.012 for SeNPs-chitosan which was considered as polydisperse, while for SeNPs-BSA the PDI was found to be 0.123 ± 0.002 which was considered monodisperse. Indeed, SeNP-chitosan had a zeta-potential of 16.4 ± 4.4mV that is within the –30 mV and +30 mV range known to indicate poor stability of the nanoparticles and very likely aggregation or agglomeration while the value of -51.2 ± 15.8 mV for SeNPs-BSA indicate a very high stability of the solution. We noticed that the average hydrodynamic size of SeNPs-BSA was 108 ± 30 nm and 320 ± 221 nm for SeNPs-chitosan this value with the ZetaSizer is higher than the supplier’s specification while the very precise measurements through series of TEM images confirmed those specifications. This can be explained by the fact that the SeNPs-solution is polydisperse and the presence of bigger particles could contribute to shift the measured particles size towards larger values through an increase light scattering.
And
Overall, the results show some differences between exposure of PC-3 cells to SeNPs-BSA or SeNPs-chitosan. The cellular uptake of SeNP-BSA is higher than that of SeNP-Chitosan; however, the IC20 of SeNP-BSA is higher than the IC20 of SeNP-Chitosan. Generally, there is a clear influence of many factors such as the cell type or route and duration of exposure concomitantly to the shape agglomeration or not and surface coating of nanoparticles. A possible explanation for the above difference in uptake could be that SeNPs-chitosan solution appears much les stable with a tendency to aggregate leading to larger particles size far from optimal for efficient intracellular uptake contrary to SeNPs-BSA. Despite much higher uptake of SeNPs-BSA the higher IC20 could be explained by the BSA coating that was reported to reduce cytotoxicity of number of type of nanoparticles, possibly preventing interactions between the reactive surface of the nanoparticles with the plasma membrane and mitigating the generation of reactive oxygen species (Fröhlich, E., 2012. The role of surface charge in cellular uptake and cytotoxicity of medical nanoparticles. International journal of nanomedicine, pp.5577-5591.)
Reviewer 3 Report
Comments and Suggestions for Authors
In this study, the authors report intracellular localization and fate of sub-toxic concentrations of SeNPs capped with BSA and Chitosan in prostate cancer PC-3 cells by various spatially resolved and speciation techniques. SeNPs displayed different intracellular uptake with SeNPs-BSA preferentially accumulated into mitochondria. In addition, SeNPs-BSA slow down the migration of PC-3 cells. Understanding the interaction of SeNP and biological systems is crucial for its anti-cancer application. Before it is accepted for publication, there are some comments should be addressed.
1. Introduction should be revised to highlight the importance of the key scientific issue studied in this paper. In addition, before introducing BSA and Chitosan(L63), it is suggested to explain why functionalization of SeNP is necessary.
2. L350-353. Besides TEM, the hydrodynamic size and PDI of NPs in water and cell culture medium should be provided to support the statements.
2) The data in Figure 2 and Table 1 do not support that BSA and Chitosan have been coated on SeNPs successfully. In addition, what’s the mass ratio of SeNP in the functionalized SeNP.
3) Please check the data of SeNP-Chitosan and sodium selecnite in Figure 7C, which are conflict to the data in Figure 7D and Figure S2.
4) Cell uptake of SeNP-BSA is higher than that of SeNP-Chitosan; however, the IC20 of SeNP-BSA is higher than that of SeNP-Chitosan. Please discuss the relationship of cell uptake, selenium speciation, and toxicity of the SeNPs.
5) A lysosome fluorescent probe was used to prove that SeNPs are in lysosomes. But, the conclusions strength that SeNPs-BSA preferentially accumulated into mitochondria. Why not used a mitochondrion fluorescent probe to prove it, as it is difficult to identify SeNPs using TEM.
6) Please keep the space between numbers and units.
Author Response

(The authors gave the same response as above.)
